# Mutations mark cell lineages and sectors in flowers of a woody angiosperm

Yilun Ji[1]ⓔ, Xiaonan Chen[1]ⓔ, Xiaohui Zhang[1]ⓔ, Wenjing Wang[1], Lan Xue[1], Yifan Zhong[1], Dacheng Tian[1], Sihai Yang[1,2], Long Wang[1]*, Milton Brian Traw[1]*, Ju Huang[3]*

1 School of Life Sciences, Nanjing University, Nanjing, China, 2 Co-Innovation Center for Sustainable Forestry in Southern China, Nanjing Forestry University, Nanjing, China, 3 State Key Laboratory for Crop Genetics and Germplasm Enhancement, Bioinformatics Center, Academy for Advanced Interdisciplinary Studies, Nanjing Agricultural University, Nanjing, China

ⓔ These authors contributed equally to this work.
* wanglong@nju.edu.cn (WL); mbtraw@nju.edu.cn (MBT); ponyhuang@njau.edu.cn (JH)

## Abstract

While radial sectors of flowers are likely to accumulate nested mutations, the distribution of natural somatic mutations across floral organs in dicot plants has not been studied previously. Here, we use next generation sequencing of 78 independent genome samples through sepals, petals, stamens, and carpel of two flowers to ask 1) whether radial sectors accumulate nested mutations, 2) whether carpels share *de novo* mutations with the radial parts, and 3) whether mutations can be used to estimate founder cell numbers in the floral anlagen. Mutations described seven sectors, each including a single petal and adjacent sepals, consistent with radial cell proliferation in the floral cup that predates the split of the sepal and petal cell populations. Mutations nested into two deep cell lineages that propagated through branching events and into every floral organ and unit of both flowers. Both carpels possessed two flower-wide *de novo* mutations, confirming that their genetic origin does not precede the floral anlagen. However, carpels possessed none of the sector marking mutations and therefore isolated genetically early in floral bud formation. Cell estimates of the flower-wide point mutations were most consistent with two cell floral initiation, one L1 and one L2, suggest a model of cell division in the floral anlagen. These observations represent the first genome-wide map of natural mutation distributions in a radially symmetric angiosperm flower.

## Author summary

Flowers represent the transition between vegetative growth and reproduction in plants. Here, we show that two peach flowers each initiated from exactly two cells that received marking mutations that propagated to all descendant cells. This is the first such demonstration for any plant. Moreover, we show that the

**Data availability statement:** The complete Xten platform -data generated during this study are available in the National Center for Biotechnology Information Sequence Read Archive repository under study accession number PRJNA1277537 which are accessible with the following link: www.ncbi.nlm.nih.gov/sra?term=PRJNA1277537. Raw data for all figures are provided (Tables S1-S13).

**Funding:** This research was funded by grants from National Key Research and Development Program of China (#2023YFD2200102 to SY and #2023YFD2200104 to LW), China National Science Foundation Grants (#32170327 to LW and #32270664 to SY), and the Fundamental Research Funds for the Central Universities (#14380209 to LW). The funders had no role in study design, data collection and analysis, decision to publish, or preparation of the manuscript.

**Competing interests:** The authors have declared that no competing interests exist.

carpel, the flower part that protects the ovules, does not share mutations with the other flower parts such as sepals, petals, or stamens, confirming that this protective structure segregates from the rest of the flower just after initiation of the flower bud.

## Introduction

Flowers come in a dazzling array of shapes, but their showy parts, typically the sepals, petals, and stamens, can be classed generally as having either bilateral symmetry (e.g., Antirrhinum etc.) or radial symmetry (e.g., roses, etc.) in the dicots [1]. There is some agreement that radial symmetry in sepals, petals, and stamens is the ancestral condition for dicots [2–4], but interestingly this does not extend to the carpel, for which bilateral symmetry is the likely ancestral condition in the eudicots [1,3–5]. Moreover, carpels develop both at a different developmental pace and with vasculature that is discrete from these other flower parts [5]. Based on these lines of evidence, the carpel is not likely to share *de novo* floral sectoring mutations with sepals, petals, or stamens, but this has not been assessed previously in any dicot flower.

Histological studies have shown that floral buds in dicot species as diverse as Arabidopsis and peach initiate from a small number of founder cells, in the range of two [6] to four cells [7]. If a flower is formed from two founder cells, then any somatic mutation that arises *de novo* in that cell will be present in all cell descendants throughout the flower. If both founder cells are marked by mutations, then cell relative abundance estimates calculated independently from pairs of the marker mutations will sum to unity in all samples of the flower parts. Natural mutations provide perhaps the only way to test this, because the mutation must arise precisely in each of the two founder cells.

Tracking of cell lineages in floral parts has been accomplished previously with sport mutations [8] and artificially induced mutations [9,10] in pigmentation genes, and more extensively with induced periclinal chimaeras, which are then tracked by nuclear staining and cell size differences [6,11]. These approaches have advantages for rapid phenotyping, but also have possible confounding effects on cell behavior [12,13]. While natural silent somatic mutations may be preferable as cell lineage markers, tracking them has involved both prohibitive expenses and computational challenges, and their utility for tracking floral cell lineages has not been shown previously for any flower.

In humans, natural mutations have been used to mark specific cell lineages and show that two such lineages persist in asymmetric abundance during tumor growth in breast cancer [14]. In these cases, mutations arise and are subfixed, defined as marking all subsequent cells from that lineage, such that the proportion of marked read abundances in the genomic samples are indicative of the cellular abundances of this cell lineage in the tissue. In the current study, we adopt these cell estimation approaches from cancer studies in combination with the next generation genomic data to present one of the first demonstrations of such a low-level constitutional mosaicism in a woody plant.

Assessment of constitutional mosaicism is possible in this model plant because both branch and floral meristems of peach retain highly conserved cellular organization, where L1 cell lineages, those forming the epidermal layer, divide only in the anticlinal direction, whereas L2 cells divide both anticlinally and periclinally [6,15]. The classic histology studies of peach cytochimeras established both the complete genetic isolation of L1 and L2 and the co-propagation of these two major lineages into all peach axial meristems. This previous work also established with rigor that the L1 layer has only one cell thickness in aerial peach parts and that the proportion of L1 derived (epidermal) cells in leaves and floral parts is low, ranging from 1:10 (leaves, sepals, carpels) to 1:4 (petals) relative to L2, from which all of the internal leaf and floral structures are derived [6,8,15]. Thus, the expected VAF (%) of a somatic mutation in L1 is not predicted to exceed 10% in leaves, sepals, or carpels or 25% in petals, under the simplest assumption of uniform ploidy in the L1 and L2 cell populations.

Whether there is a genetic lottery among sublineages within L1 or L2 in terms of who populates the new axillary meristems is an interesting unanswered question. If there are multiple L1 cell sublineages present in each meristem, then the VAF (%) values for a particular L1 somatic mutation could either be fractionally lower in distal tissues (if each L1 sublineage is propagated into each axial meristem, i.e., the sublineages are conserved) or the max L1 value or nothing (if only one L1 sublineage is propagated into each axial meristem, i.e., the sublineage diversity is replaced by the single successful sublineage). Recent studies have indicated evidence for both models of cell propagation depending upon the tree species considered [16–18].

Flowers in peach (Fig 1) and other members of the rose family (Rosaceae) typically exhibit five-parted radial symmetry, with five sepals, five petals, 30–40 stamens, and a single carpel [6]. Histological studies have demonstrated that the central carpel exhibits a distinctly isolated position, whereas sepals, petals, and stamens share close physical proximity to each other on the floral cup and share vascular bundles that are distinct from those leading to the carpel [19–21]. Sepals in peach are coterminous around the full perimeter of the floral cup (Fig 1A), whereas petal attachments are discrete and pedunculate, emerging at the sepal junctures, and occupying a total of a third or less of the perimeter of the floral cup (S1 Fig). This leads to the prediction that sepals will capture more deep mutation events than petals and that mutations shared by two sepals will be present in the intervening petal if that petal arises from the same primordial cell group. To date these predictions have not been evaluated.

Here, we use genome-wide tracking of natural *de novo* mutations in two peach flowers, hereafter "Flower #1 and #2, to ask 1) whether shared mutations describe radial sectors that unite sepals or sepals and intervening petals, 2) whether carpels share *de novo* mutations with the radial parts, and 3) whether mutations can be used to estimate founder cell numbers in the floral anlagen. During processing, we split twenty samples from Flower #2 and obtained independent genome calls on both subsamples to quantify technical variation in point mutation estimation [22].

## Results

### Distribution of point mutations

To assess mutation distribution and abundance in sepals, petals, stamens, and carpels, we obtained 78 genomic samples from two flowers (Fig 1B), of which 20 samples were then split to obtain paired independently genome-sequenced subsamples, for a total of 98 genomic measurements of the flowers (S2 Fig). For comparison, we also measured ten genomic samples from one subtending leaf of Flower #2, for a grand total of 108 genomic measurements analyzed by Illumina Xten platform.

To identify somatic point mutations (S3 Fig), we mapped to the v2.0 peach reference genome [23]. The proportion of high-quality bases (>20) exceeded 92% with an average of 64.3x clean read depth (S1 Table). We created assemblies with BWA-mem 0.7.10-r789, sorted and removed PCR repeats with MarkDuplicates in Picard (v1.114), and locally reassembled with IndelRealigner in RealignerTargetCreator (S3C Fig). To extract putative single nucleotide polymorphic sites, we used Haplotype Caller in GATK 4.0 and UnifiedGenotyper in GATK3.5 (S3D Fig), and only sites with a patch assembly mass greater than 20 (error rate 1%) were included. This resulted in a preliminary set of 1,888,276 SNV for filtering.

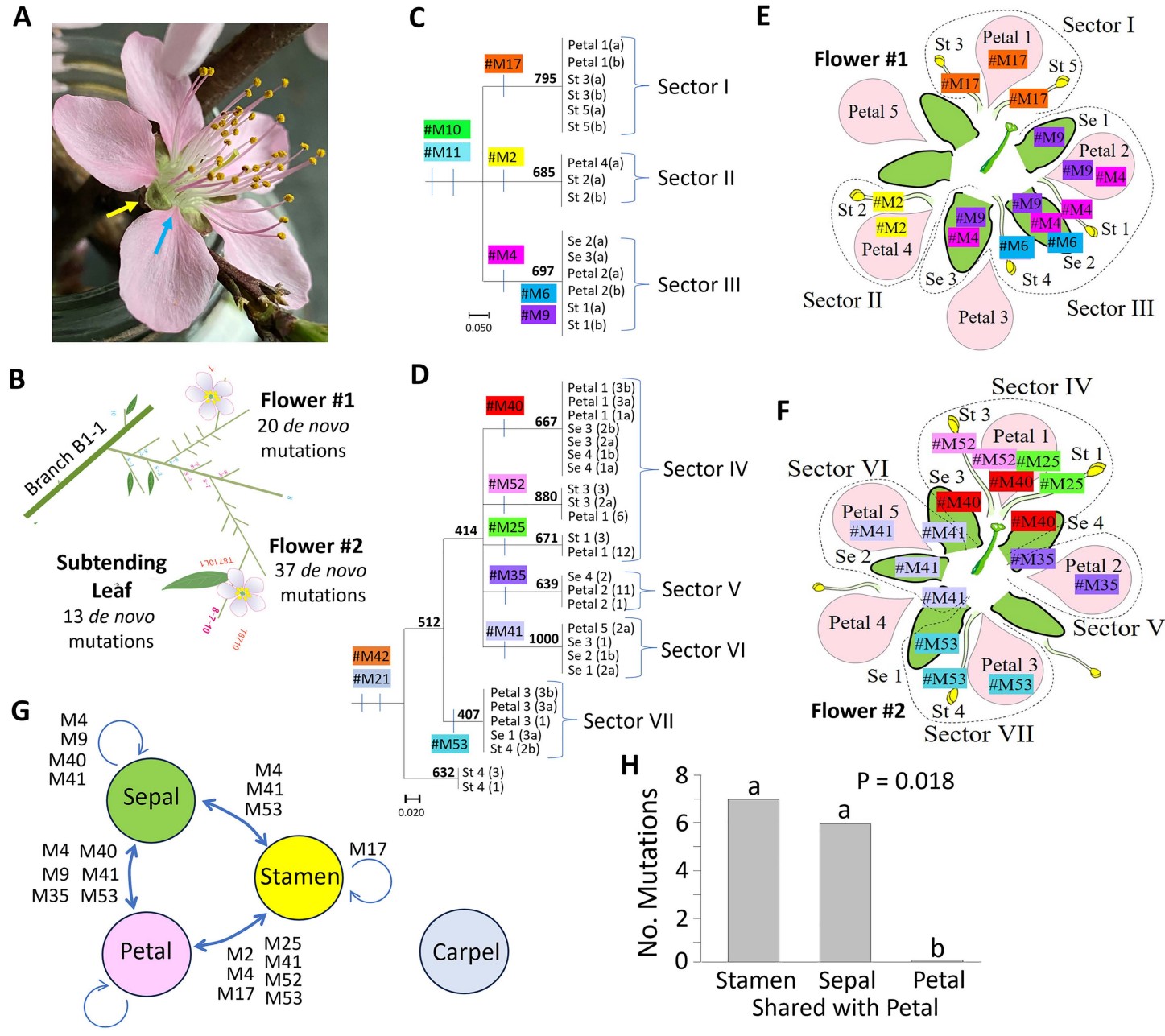

**Fig 1. *De novo* mutation distributions in two peach flowers.** A) Peach flower showing petal attachment (blue arrow) spaced by intervening sepal (yellow arrow, S1 Fig). (Photo credit: B. Traw). B) Locations of Flower #1 and #2 on sub-branches B1-1-8-4-7 and B1-1-8-7-10, respectively, C, D) Dendrograms from clustering analysis identifying C) three floral sectors in Flower #1 (Sectors I-III) and D) four floral sectors in Flower #2 (Sectors IV-VII). Bootstrap support shown (N = 1000). K-means clustering and raw data provided (S4 Table). E, F) Sectors mapped onto floral units of E) Flower #1 and F) Flower #2. G) Arrows showing mutations shared between sepals, petals, and stamens, but none shared with carpels. Mutations shared by two or more sepals or two or more stamens shown by circular arrow. H) Bar plot showing petal mutations shared with stamens and with sepals, but not with other petals. P value from Chi-square test (P = 0.018, $\chi^2$ = 6.61, df = 2).

Progressively, our filters (S3E Fig) removed SNVs representing parental heterozygosity, followed by SNV regions with poor quality (poor read quality scores, three variants at SNV, strand bias, or INDELs) and finally visual inspection in IGV viewer. SNPs present in the controls or otherwise did not pass visual inspection or were present in controls. Our controls included 1) samples from the other flower and 2) a published whole-canopy collection of leaf samples from the four main branches of the tree [24]. A *de novo* floral mutation is thus defined here as being present in that flower but absent from all other samples from the tree. Through these filters, we identified 70 *de novo* point mutations (S2 and S3 Tables), of which 20 were in Flower #1 (denoted #M1-20), 37 in Flower #2 (#M21-57), and 13 in the subtending leaf to Flower #2 (denoted #M58-70). These 70 *de novo* point mutations were entirely absent from a comprehensive whole canopy inventory of *de novo* point mutations from all 75 leaves (#M71-#M120) produced by the tree in a previous year (S2 Table) [24]. Moreover, the 57 *de novo* floral mutations were not recovered in any of 28 leaves collected distally to either flower on the branch in two subsequent years, indicating that these mutations were specific to the two flowers. Across the eight chromosomes, the mutations were distributed normally and were separated by an average distance of 1.7Mb (S4 Fig).

### Sectoring mutations cluster sepals, petals, and stamens apart from carpel

Of the 57 point mutations identified in the flowers, 15 occurred in multiple floral parts (S5 Fig) and were therefore potentially informative for sectoring. These 15 mutations identified three sectors in Flower #1 (Fig 1C and 1E) and four sectors in Flower #2 (Fig 1D and 1F), each containing exactly one petal. These seven sectors had bootstrap support exceeding 60% (except Sector #7 with 40% support) and matched the output of K-means clustering (S4 Table). Six of the 15 mutations (#M4, #M9, #M35, #M40, #M41, and #M53) were shared between a sepal and a petal, indicating that these six mutations were in cells that preceded the split of the sepal and petal cell populations (Fig 1G). Four of these six mutations (#M4, #M9, #M40 and #M41) linked two or three adjacent sepals, indicating that these four mutations occurred prior to the isolation of sepal cell groups from each other. These same four mutations were all found in the intervening petal as well (Fig 1E and 1F), indicating that the petal and sepals were derived in part from shared cell populations. In contrast, no mutations were shared between petals ($\chi^2 = 6.61$, df = 2, P = 0.018, Fig 1H). Thus, adjacent sepals shared marked ancestral cell populations, whereas adjacent petals did not.

### Qualitative and quantitative repeatability in split samples

To assess technical repeatability of mutation calls, we compared output from each of the 20 pairs of split subsamples collected from Flower #2, measuring both qualitative and quantitative consistency (Fig 2). Qualitatively, we found that allele variant calls in paired subsamples matched in 97.1% (719 of 740) of possible comparisons, where 740 is the product of the 37 *de novo* mutation sites in Flower #2 assessed in each of the 20 split sample pairs (S6 Fig and S5 Table). Of greater importance for the following assessment of mutation nesting is repeatability of quantitative estimates of mutation marked cell abundances in different samples. For quantitative assessment, we used read counts in the paired subsamples to assess the percent variant allele frequency (VAF%), calculated as (100*(#ALT reads/total reads) using all 150mer reads that overlay the base position of that point mutation (Fig 2A and S6 Table). We found that the paired subsamples had highly significant positive correlations in VAF (%), as shown for #M42 ($R^2 = 36.7$, P < 0.001; Fig 2B), which we present because it was the most widespread *de novo* floral mutation in Flower #2 and present in all 20 split samples and with the combined data for all 37 *de novo* mutations of Flower #2 together ($R^2 = 79.5$, P < 0.0001; Fig 2C). Collectively, these data from the split samples indicated strong confidence in repeatability of both qualitative and quantitative differences.

### Mutations nest into two deep branch lineages

If mutations at meristem initiation are rare, we would not expect to see any flower-wide *de novo* mutations, given that only two focal flowers were being studied. To our surprise, there were four such flower-wide *de novo* mutations, namely #M10 and #M11 in Flower #1 (Fig 2D) and #M42 and #M21 in Flower #2 (Fig 2E). Moreover, the VAF abundances of these

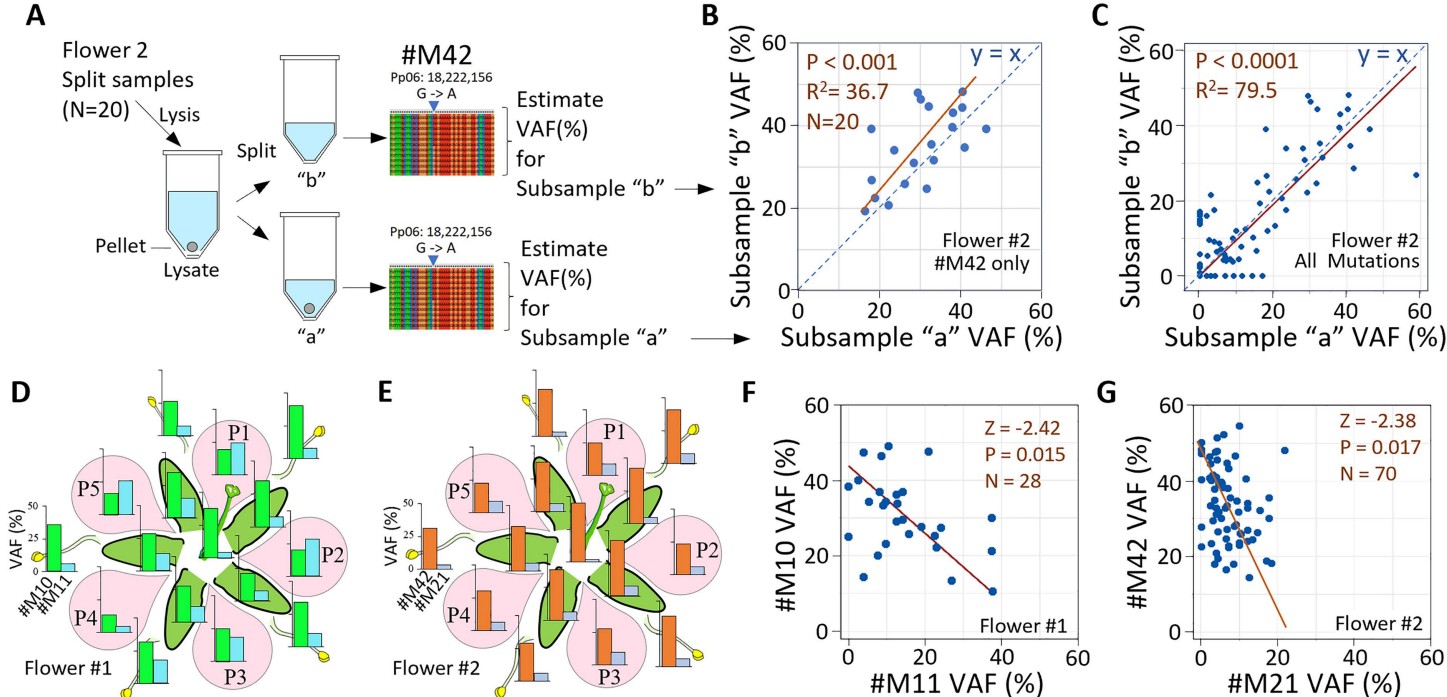

**Fig 2. VAF quantitative check and distributions of flower-wide *de novo* mutations.** A) Diagram showing lysate partitioned into two subsamples for each of twenty split samples in Flower #2. Each subsample was genome sequenced independently and assessed for mutations and variant allele frequency (VAF) following standard protocol with VAF (%) calculated as 100*marked reads/total reads at that base position. B, C) VAF (%) estimates for subsamples "a" and "b" correspond positively across the twenty split sample pairs for B) focal mutation #M42, the only *de novo* mutation present in all subsamples, and C) all 37 *de novo* Flower #2 mutations combined. Raw data provided (S6 Table). Orthogonal regression line, significance, and sample size (brown) and y = x line (blue dashed line) shown. D, E) Bar plots of flower-wide *de novo* mutations D) #M10 (green) and #M11 (blue) in Flower #1 and E) #M42 (orange) and #M21 (navy) in Flower #2. Scale bars show VAF from 0 to 50% in all plots. F, G) Scatterplots showing negative correspondence between VAF(%) of F) #M10 and #M11 in Flower #1 and G) #M42 and #M21 in Flower #2. Orthogonal regression, significance, and sample sizes shown (brown). Raw data included (S3 and S6 Tables).

flower-wide *de novo* mutations were negatively correlated with each other, as seen in the scatterplots for #M10 and #M11 (Fig 2F) and #M42 and #M21 (Fig 2G). Samples in Flower #1 with elevated levels of #M10 had low levels of #M11 and vice-versa, while samples in Flower #2 with high #M42 had low levels of #M21, and vice-versa (S7 Fig). The classic histological observations with cytochimeric mutants were conclusive in showing that there are only two main underlying cell lineages, one with low abundance and the other with high abundance, in peach flowers [6,15], but whether such lineages could accumulate nested mutations has not been studied previously. Having found these pairs of negatively correlated low and high VAF (%) mutations in each flower, we wondered if the two low abundance flower-wide floral mutations (#M11 and #M21) could be marking the same deep low abundance branch cell lineage and whether the two high abundance flower-wide mutations (#M10 and #M42) could be marking the other deep high abundance branch cell lineage, previously identified by Dermen and colleagues.

To assess whether the floral mutations nested within deep lineages within the main branch (Fig 3), we correlated the VAF (%) of these *de novo* mutations in the floral samples to a previous published set of branch mutations from a whole canopy survey of all 75 individual leaves from this same tree DHQ1 two years earlier, where 48 *de novo* point mutations were identified in the four main branches [24]. Scanning the known branch mutations (S2 Table), we found that VAF (%) of two of the deepest branch mutations, #M101 and #M119 (Fig 3A), were highly positively correlated with each other (Z = 9.69, P < 0.00001, Fig 3B), and both were correlated positively with #M11 (Z = 3.88, P < 0.001, Fig 3C) and #M21 (Z = 3.23,

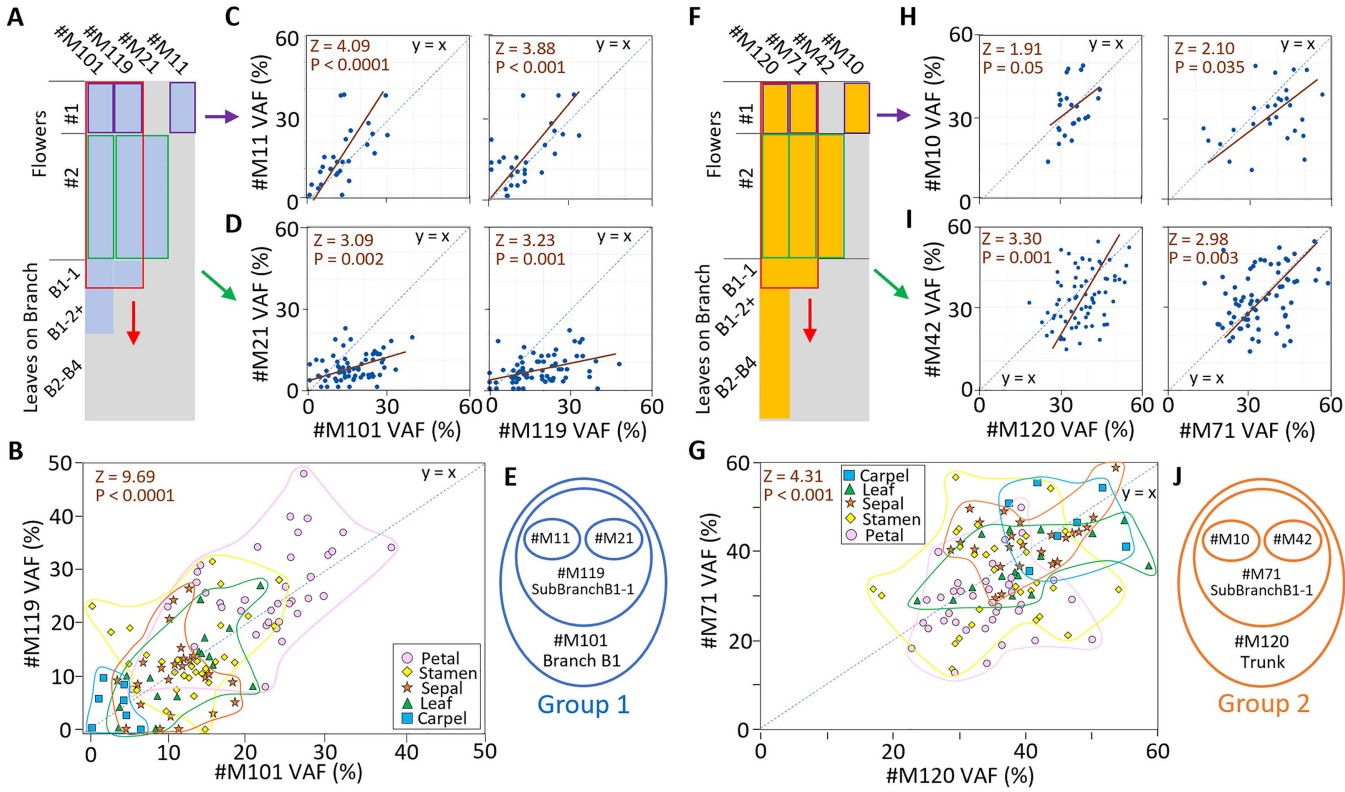

**Fig 3. Floral *de novo* mutations nest within two deep branch cell lineages.** A-E) Low abundance cell lineage hereafter "Group 1" marked by four nested mutations consisting of #M11 in Flower #1, #M21 in Flower #2, #M119 in the sub-branch and #M101 in the main branch. A) Presence (blue) and absence (gray) of Group 1 mutation markers for samples for Flower #1 (purple), Flower #2 (green), and both flowers and branch B1-1 leaves (red). Positive VAF (%) correlations of B) #M119 with #M101, C) #M11 with #M101 and #M119, and D) #M21 with #M101 and #M119. E) Venn diagram summarizing nesting in Group 1 mutations. F-J) High abundance cell lineage hereafter "Group 2" marked by four nested mutations consisting of #M10 in Flower #1, #M42 in Flower #2, #M71 in the sub-branch and #M120 in all branches. F) Presence (orange) and absence (gray) of Group 2 mutation markers. Positive VAF (%) correlations of G) #M71 with #M120, H) #M10 with #M120 and #M71, and I) #M42 with #M120 and #M71. J) Venn diagram summarizing nesting in Group 2 mutations. Five outliers excluded from scatterplot of #M10 and #M120. All raw data, permutation tests, and orthogonal regression fits provided (S8 Fig and S7 Table).

P = 0.001, Fig 3D), with the four nested mutations together marking a low abundance cell lineage "Group 1" (Fig 3E). Looking again at the known branch mutations (S2 Table), we found that the only other two deep mutations, #M71 and #M120 (Fig 3F), were correlated positively with each other (Z = 4.33, P < 0.001, Fig 3G), and correlated positively with #M10 (both Z = 2.1, P = 0.035, Fig 3H) and #M42 (Z = 2.98, P = 0.003, Fig 3I), thus marking a high abundance cell lineage "Group 2" (Fig 3J). Mutation read proportions of the Group 1 and Group 2 markers estimated cell variant frequencies (CVF) for which the sums approached 100% for four marker pairs (S9 Fig and S8 Table), and matched quantified cell abundances of L1 and L2 in floral and leaf tissues (S10 and S11 Figs and S9 Table) and sampling of epidermal tissues (S12A–S12E Fig and S10 Table). Group 2 markers were found in progeny, whereas Group 1 markers were not (S12F Fig and S11 Table). Collectively, these two independent lines of evidence indicated Group 1 and Group 2 as the likely L1 and L2 lineages, respectively.

## Mutations suggest single founder cells in L1 and L2 of the anlagen

Dermen and colleagues showed that each peach floral meristem initiates with at least one L1 (Epidermis) founder and one L2 (Inner tunica) founder, and no specifiable representative of L3, but could not establish whether these were single cell

origins or included multiple priors of each type [6]. This can be established however with *de novo* point mutations if they occur during formation of the floral anlagen. If the flower L1 layer is initiated with a single cell, then a somatic mutation that arises during the mitotic event leading to that founder cell should be passed on to every single L1 cell of that mature flower. Likewise, if the flower L2 layer initiated with a single cell, then a mutation arising during the mitotic event leading to that founder with result in the interior cells of the flower all containing that mutation as well.

This is what we observe with three of the four flower-wide *de novo* mutations (Fig 4). Mutations #M42, #M11 and #M10 all exhibit VAF (%) abundances (Fig 4A–4D) that are fully on par with the VAF (%) values of the deep branch mutations (Fig 4E), thus the data are most consistent with each of these three floral mutations occurred in a single cell prior in the

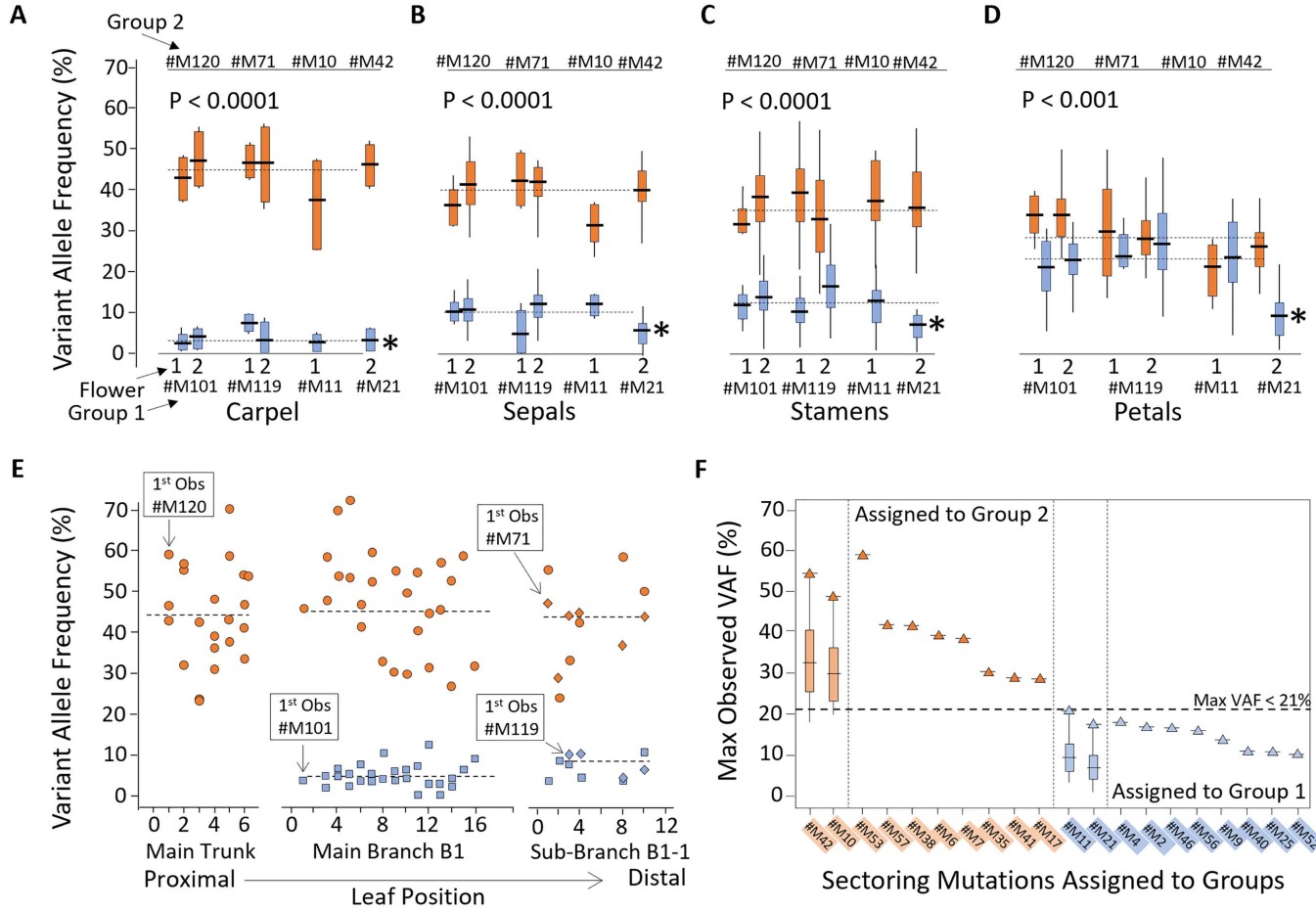

**Fig 4. Both deep cell lineages retain differential VAF (%) across floral parts and leaves.** A-D) Box and whisker plots shown for VAF (%) of #M101, #M119, #M11, #M21 (Group 1) and #M120, #M71, #M10, #M42 (Group 2) in both flowers for A) Carpel, B) Sepals, C) Stamens, and D) Petals. Mutation #M21 (asterisk) exhibited half abundance of the other Group 1 mutations, suggesting origin one cell division later. P value from one-way ANOVA. Raw data provided (S7 Table). E) Scatterplot of VAF (%) in branch leaves as a function of leaf position (trunk to distal tip of Branch B1-1) showing that deep mutations #M120, #M71, #M101, and #M119 were already at or above their averages (hashed lines) at first observation. F) Eight sectoring mutations assigned to Group 2 (#M53, #M57, #M38, #M6, #M7, #M35, #M41, and #M17) and eight assigned to Group 1 (#M4, #M2, #M46, #M56, #M9, #M40, #M25, #M52), respectively. Assignment based on maximum observed VAF (%) values (triangles) in carpel, sepals, or stamens, but not petals. Mutations ordered from highest to lowest max VAF(%). Threshold determined as max VAF (%) > 21.0% for Group 2 inclusion and max VAF(%) < 21.0% for Group 1 inclusion, where 21.0% was the maximum value observed for the two *de novo* flower-wide Group 1 mutations (#M11, #M21) in carpels, sepals, or stamens. Box and whisker plots included for the four flower-wide mutations, #M42, #M10, #M11, and #M21. Mutation values from petal samples were excluded due to the overlapping distributions (shown in Panel D). One outlier for #M11 in a stamen excluded. Raw data provided (S12 Table).

establishment of the anlagen. In contrast, #M21 was consistently at half abundance relative to the other Group 2 mutations (denoted by asterisk in Fig 4A–4D), suggesting that it arose in the first daughter of an unmarked founder cell. Importantly, none of these four flower-wide *de novo* floral mutations (#M42, #M21, #M11, or #M10) were recovered from any of 14 leaves collected distally on either floral sub-branch (Sub-branches B1-1-8-4-7 and B1-1-8-7-10) in the two subsequent growing seasons.

To further assess the tempo of the mutation events, we plotted the VAF(%) of the four major branch mutations (#M120, #M119, #M71, #M101) in leaves along the branches and found that they achieved peak cellular abundances at first observance and typically just after a branch meristem initiation event (Fig 4E and S7 Table). This suggested that branching events also represented a stringent cellular bottleneck, supporting previous branch mutation patterns from *Salix* [25], *Eucalyptus* [26], and *Osmanthus* [27]. While nearly all *de novo* mutations reported in the current study are in intergenic regions (S2 Table), #M101 is an exception in that it is a truncation mutation in the final exon of a mudrA mutator family gene.

## Mutations mapped onto the floral meristem cross-sections

We returned to the 15 floral sectoring mutations and assigned them to these major lineages (Fig 4F and S12 Table) based on their highest observed VAF (%) value, with a threshold of VAF > 21% for Group 2 and VAF < 21% for Group 1. Mutations that only occurred in petals and mutations with two or fewer records could not be assigned. We then mapped cell lineage progression for both major lineages through both flowers (Fig 5). In Flower #1, Group 1 fixed mutation #M11 in the anlagen, then picked up #M2 in Sector II and #M4 and #M9 in Sector III (Fig 5A), whereas Group 2 fixed mutation #M10 in the anlagen, then picked up #M17 in Sector I and #M6 in Sector III (Fig 5B). In Flower #2, Group 1 fixed #M21 in the anlagen, then #M40, #M25, and #M52 in Sector IV (Fig 5C), whereas Group 2 fixed #M42 in the anlagen, then picked up #M53 in Sector VII (Fig 5C), #M41 in Sector VI, and #M35 in Sector V (Fig 5D). These cross-sections show that mutations mostly identified petal/ stamen pairs, for which the split of the primordial cell population occurs later in the formation of the floral cup (S13 Fig).

## Discussion

A flower begins as a swelling initiated on the shoulder of the shoot apical meristem consisting of a small number of founding cells [7,28–30]. Histological studies with cytochimeric plants showed that this initial population in peach must include at least one L1 founder and one L2 founder [6], but could not rule out whether additional cells of each lineage pass into the new structure as well. Here, we tracked natural mutations through peach flowers and found evidence that both studied flowers developed from exactly two cells. This is the first demonstration of natural mutations in founder cells in an angiosperm flower. Such identification of a two cell bottleneck at floral initiation will be important information for stochastic estimation of the accumulation of mutations in plant lineages between generations [31,32].

Mutations also described seven sectors, each including a single petal and adjacent sepals, consistent with radial patterning of cell proliferation in the floral cup that predates the split of the sepal and petal primordial cell populations. Carpels did not possess any of the sector marking mutations suggesting that carpel cell lineages were isolated early in floral bud formation. Mutations nested into two deep cell lineages that propagated through branching events and into every floral organ and unit of both flowers. The ratio of these two deep cell lineages match closely to the proportions of L1 and L2 cells in histological sections through the tissues. This is the first such demonstration of nested mutations in sectors in any angiosperm flower.

Recent studies have indicated evidence for two different models of cell propagation (conserved versus replaced) depending upon the tree species considered [16–18]. If there are multiple L1 cell sublineages present in each meristem, then the VAF (%) values for a particular L1 somatic mutation could either be fractionally lower in distal tissues (if each L1 sublineage is propagated into each axial meristem, i.e., the sublineages are conserved) or the max L1 value

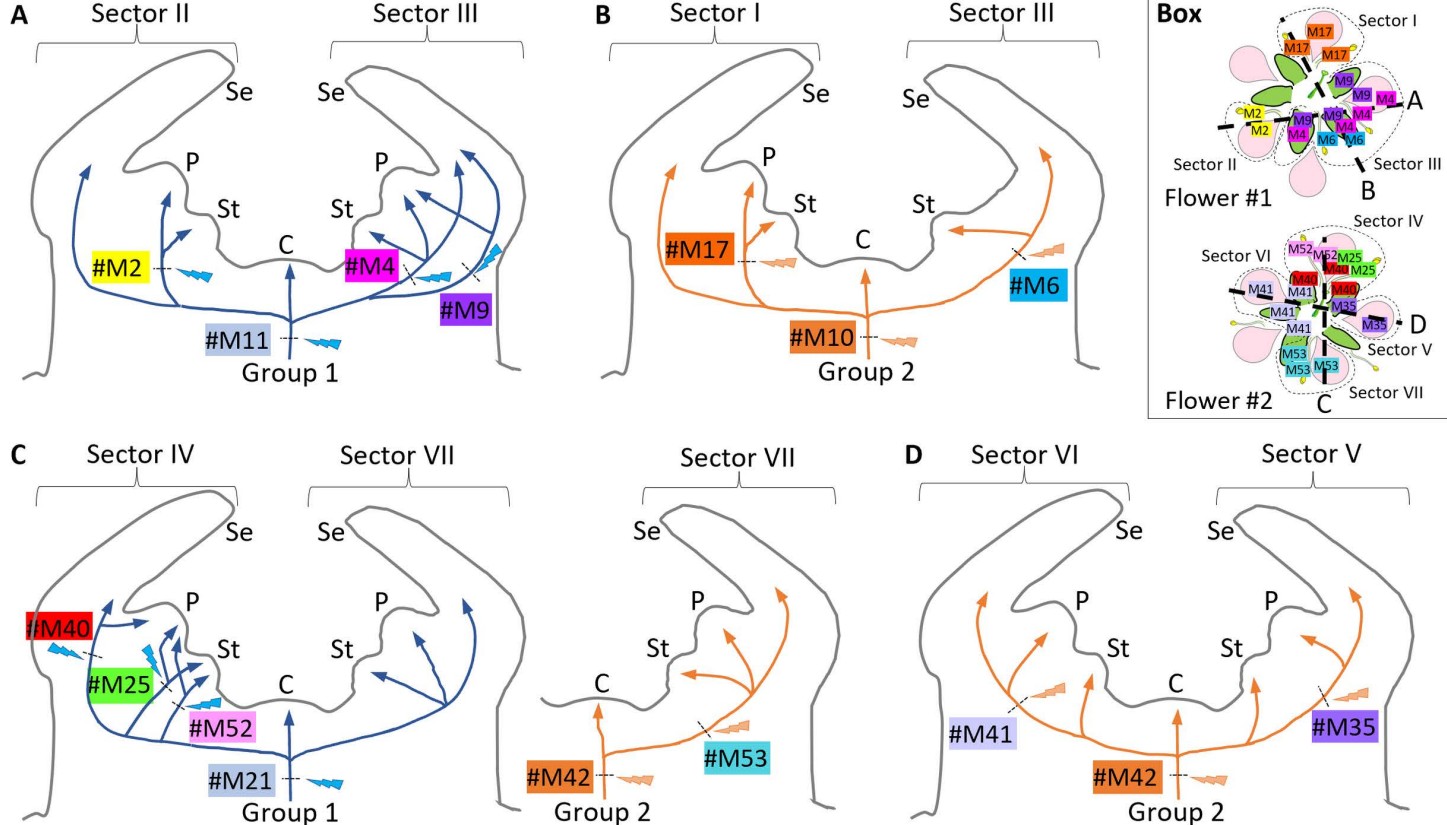

**Fig 5. Floral cross-sections with mapped *de novo* mutations in sectors.** A, B) Two cross-sections through Flower #1 showing A) Sectors II and III as marked by mutations in Group 1 cell lineage and B) Sectors I and III as marked by mutations in Group 2 cell lineage. C, D) Two cross-sections through Flower #2 showing C) Sectors IV and VII as marked by mutations in the Group 1 and Group 2 cell lineages and D) Sectors VI and V as marked by mutations in Group 2 cell lineage. Inset box shows locations of the two cross-sections through each of the two flowers. Cross-section histology provided (S13 Fig).

or nothing (if only one L1 sublineage is propagated into each axial meristem, i.e., the sublineage diversity is replaced by the single successful sublineage). Here in peach (Fig 4E), the pattern appears more consistent with replacement where first the unmarked L1 was replaced by the #M101 marked L1, and then subsequently that was in turn replaced by the #M101 + #M119 marked L1. While these two mutations do not support a genetic lottery among sublineages of L1 across the branches studied, more sampling is needed to establish which pattern is more common in peach and other angiosperms.

Stewart and Dermen (1970) recognized rare instances of exchange between L1 and L2, which they termed replacement if L1 cells divide downward and become the top layer of L2 or even entirely replace L2, and displacement, if L2 cells take over the L1 role, as can occur if the L1 layer is damaged [13]. However, they emphasized the "surprising stability of cellular position at the center of growth" for both L1 and L2 and supported this with an example where they followed marked cell lineages for over one hundred nodes without change. In our samples, if L1 replaced L2 in the branches or flower, we would see this as a dramatic increase in the proportion of Group 1 cells in a distal sample. However, despite extensive sampling in this study, we never observed a Group 1 mutation value exceeding 55% of the cell population, thus suggesting no instances where L1 cells divided periclinally and thereby replaced the existing L2 cells. Conversely, if L2 displaced L1 in our samples, we would see samples lacking all Group 1 markers. In our dataset, there were rare samples

missing one Group 1 marker (S7 Table), but no samples where all Group 1 markers were absent, indicating that displacement did not occur. Thus, one of the most useful aspects of our next gen sequencing work presented here is likely to be the recovery and independent confirmation of the two deep genetically distinct cell lineages in peach, which were first demonstrated with rigor over 50 years ago by Dermen and colleagues in their work with cytochimeric peach lines.

Mutation rates for peach should be estimated separately for L1 and L2 in future studies. This is important for several reasons. First, only L2 contributes to offspring formation in peach [6] and thus only mutations in L2 are therefore likely to be passed on to progeny. Using bulk collected whole branch level mutations to estimate somatic mutations per generation would grossly overestimate the true transgenerational rate in peach because such a set would include all of the L1 mutations. L1 mutations must be identified and set aside from such mutation per generation calculations in this species. Second, much interesting information will likely be gleaned by looking separately at somatic mutation rates in L1 and L2. L1 is physically on top of L2, i.e., distal to L2 from the center of the plant, and thus predicted to have greater exposure to environmental mutagens. For this reason, observed differences between L1 and L2 in their mutation rates may well yield new insights into these environmental mutagens, may help explain ecological interactions with other species, reviewed in [33], may inform latitudinal gradient patterns or the lack thereof [34], or possibly explain unusual tree architectures (e.g., stem fruiting, etc.). In the current work, we show that it is possible to assign mutations to L1 and L2 based on observed maximum VAF (%) values with fairly high confidence (Fig 4F). This approach is likely to be useful also for future studies on other woody tree species, to the extent that distinct L1 and L2 genetic lineages are more likely the rule than the exception in angiosperms [35]. That said, the number of fully nested mutations in each lineage shown here totals three, which is not a sufficient number of events for rate calculations. In upcoming work, we will address mutation rates for both L1 and L2 in peach using a larger set of mutations collected from more recent branch and progeny samples for this tree.

Two studies have recently been published, one on apricot fruit [35] and the other in potato vegetative tissues [36], that also partition somatic mutations in L1 and L2 tissues. Of the two studies, the one on apricot is closest, given that it is on a woody plant, but differs from the current study in that it includes both point mutations and insertion/deletions [35]. Restricting their data to point mutations, we find excellent correspondence in the observed frequency of mutation events, where we observe two nested L1 and one nested L2 point mutations between the two flowering branches, while they observe approximately ten nested L1 and five nested L2 point mutations across seven fruiting branches. Thus, independently, our peach and their apricot datasets both suggest exceedingly low rates of point mutation in woody tissues of *Prunus*, on the order of one event per cell lineage per branch initiation. This estimate is in line with the findings of earlier studies in other woody trees [26,27,37–39]. However, these studies were all conducted on leaves. With respect to other studies, our mutation numbers per biological sample reported here for flowers (~0.7) are at the low end of the range observed in other studies of woody angiosperm tissues, where the average is around 4 mutations per sample, with substantial variation among studies (S13 Table). Notably, none of the prior studies have explored differences among floral parts. Our stringent filtering of candidates in the mutation identification pipeline may also contribute to the low estimate. Future studies should not filter candidates by requiring a minimum read copy number per mutation, given that some mutations may be filtered out through the use of such a threshold (S13 Table), as recently suggested [40].

Endopolyploidy is a broadly observed phenomenon across dicots, occurring most frequently in specialized cell types, such as petal pigment cells, trichomes, epidermal pavement cells, and near vascular bundles in leaves [41]. Such cells are typically unable to undergo subsequent additional cell divisions, but the high ploidy in such cells (up to 128C or more) could theoretically explain discrepancies between histological counts of cell frequencies and next-gen read count-based estimates of CVF (%). To explore this possibility, we compared three scenarios (S14 Fig), with either no endopolyploidy, endopolyploidy in L1, or endopolyploidy in both L1 and L2. This is expected because in most plants both L1 and L2 produce endopolyploid cells [41] and thus the read contributions from L1 and L2 cells in a genomic sample retain the ratios of the underlying cell proportions (Scenario #3). Cell percentages calculated independently from markers in each of the two lineages summed to near 100% (S9 Fig). Petal samples were the only group in our study (Fig 4D) to consistently deviate

from the L1:L2 proportions in leaves and this might be explainable if the petal epidermis in peach has an elevated rate of endopolyploidy, given that even a small amount of additional endopolyploidy in L1 would cause overestimation of cell numbers of that lineage (Scenario #2). Additional data on rates of endopolyploidy in floral tissues may be helpful in future studies.

While most *de novo* mutations were in intergenic regions, it is intriguing that one of the mutations (#M101) is a truncation mutation in the final exon of a mudrA mutator family gene (S2 Table). This mutation is close to but outside of the MUG family of Arabidopsis [42], and in a relatively unknown group of genes with both monocot and dicot homologs. Interestingly, one of the closest homologs is the TED gene of maize which has been observed to exhibit similar truncation mutations specifically in somatic cells [43]. Whether mudrA genes are specifically involved in cell lineage identity in either L1 or L2 lineages remains unknown.

In the medical literature from human cancer studies, mutation marker tracking has shown that cell lineages persist in asymmetric abundances in breast cancers [14] and during normal human embryonic development [44,45]. In these cases, mutations arise and are subfixed, such that their marked read proportion abundances in the genomic samples are indicative of the cellular abundances of this cell lineage in the tissue. If all the lineages in the tissue are marked by these mutations, then the combined sums of the mutation marked estimated proportions will approach unity (100%) when sampled across independent time points and anatomical locations. In the current study, we have mutations that appear to have subfixed in the epidermis (L1) and other mutations that subfixed in the inner tunica (L2). These data are among the first to show that quantitative cell lineage estimation in woody plants is feasible from next generation sequencing data.

In summary, we have reported for the flowers studied here that: 1) radial sectors accumulated nested mutations shared by sepals but not by petals, 2) carpels did not share *de novo* sectoring mutations with the radial parts, and 3) both flowers arose from exactly two founder cells to form the floral anlagen. These are the first such assessments of whole genome *de novo* mutation distributions in an angiosperm flower. Future studies of flowers in other species will establish whether such patterns are indicative of general conditions in the angiosperms.

## Methods

### Plant material

Cultivated peach (*Prunus persica*) is a woody tree in the Rosaceae with large radially symmetric five parted flowers and a well-annotated diploid genome (2n = 16) of compact size (227Mb) [23]. It was one of the first woody plants sequenced [23] and has been a model for floral study [8,46] and mutation tracking [24,47,48]. We focus here on tree DHQ1 which we germinated from seed in 2015 and sampled previously for mutations in 75 leaves collected in 2016 [24]. Sub-branch 1-1-8 appeared in 2017 and then the two sub-branches (1-1-8-4-7 and 1-1-8-7-10) both emerged in June - July, 2018. The two flowers (1-1-8-4-7-T) and 1-1-8-7-10-T), referred to as Flowers #1 and #2, were collected in May, 2019. Counting from the main trunk, these flowers shared four branch meristem initiation events in common and counted as node events from the trunk were at the 21st and 27th nodes, respectively. Floral structure in this tree is simple and consists of five sepals, five petals, 30 or more stamens, and a single carpel (S1 Fig). Here, we assess these flowers with a total of 78 microsamples (S2 Table). We included the five sepals, five petals, carpel, and a subset of five stamens for each flower. Twenty samples from Flower #2 were split into two subsamples each for the repeatability assessment, for a total of 98 independent genome estimates from the flowers. In addition, we assess ten microsamples from the nearest subtending leaf (1-1-8-7-10-L1) of Flower #2. Thus, the dataset includes 108 separate genome calls in the leaf and two flowers (S2 Table).

### DNA extraction and amplification

Tissues were collected from sepals and petals by Harris micro-punch (diameter 0.5mm) and from stamens and carpel by razor blade (S2 Fig). To avoid cross-contamination, we used a new plunger and receptacle of the punch or new razor-blade for each sample. Stamen samples consisted of the filament only in order to focus on the mitotically derived tissue.

Samples were stored in phosphate buffered saline solution. To minimize temperature-induced mutations, we treated samples with 2.5 μl alkaline lysate (400mM KOH, 100mM DTT, 10mM EDTA) for 10min on ice, then stopped with 2.5 μl neutral reaction fluid (4400mM HCl, 600mM TriCl). Otherwise, cell lysis and DNA amplification followed protocol for the Qiagen REPLI-g single-cell kit.

## Sequencing and mutation screening

Sequencing and data processing steps followed standard approaches for point mutation identification (S3 Fig). Amplification products were sequenced by Illumina Xten platform (BGI-Shenzhen) and mapped to the v2.0 peach reference genome [23]. The proportion of high-quality bases (>20) exceeded 92% with an average clean read depth of 64.3x (S1 Table). We created the assemblies with BWA-mem 0.7.10-r789, sorted and removed PCR repeats with MarkDuplicates in Picard (v1.114), and locally reassembled with IndelRealigner in RealignerTargetCreator (S3C Fig). To identify *de novo* mutations that arose specifically in each flower, we used two main types of controls. These controls included 1) samples from the other flower and 2) a published whole-canopy collection of leaf samples from the four main branches of the tree [24]. A *de novo* floral mutation is thus defined as being present in that flower but absent from all other samples from the tree.

To extract putative single nucleotide polymorphic sites, we used the current standard Haplotype Caller in GATK 4.0 in combination with earlier use of UnifiedGenotyper in GATK3.5 (S3D Fig), and only sites with a patch assembly mass greater than 20 (error rate 1%) were included. This resulted in a preliminary set of 1,888,276 SNV for filtering. The filtering pipeline for these SNV (S3E Fig), consisted of the follow three main steps to remove: 1) SNVs representing parental heterozygosity, 2) local SNV regions with poor quality (poor read quality scores, three variants at SNV, strand bias, or SNV next to an INDEL) and 3) SNPs present in the controls or otherwise did not pass visual inspection. To remove parental variants, we remove sites with more than 2 alt reads found in any control (--max-cmp-depth 2) and removed sites with alt reads observed in more than 3 controls (--max-cmp-total 3). This filter removed 1,848,688 SNVs, leaving 39,588 SNV that were not due to parental differences. To remove sites with poor local quality, we filtered out sites with no sample with at least 5 reads (--min-supp-depth 5), missing in more than three controls (--max-cmp-miss 3), strand bias, or exhibiting strand bias. To remove SNPs that were adjacent to INDELs, we matched the SNVs and INDEL positions and removed SNVs with overlap. This filter removed 39,409 SNV, leaving 179 SNV with good general local site quality. Of these 179 SNPs, 109 occurred in SNV site clusters or exhibited poor overall read alignment and were excluded, leaving 70 high quality candidate *de novo* mutations in the flowers.

We then reprocessed the full set of variant candidate files using the currently standard approach of HaplotypeCaller in GATK (4.2.4.0) using the GVCF mode and merged the individual gvcfs into a vcf file containing only variant sites. Here, we filtered out unreliable SNVs using GATK VariantFiltration with the following filters: QD (Qual By Depth)<2.0, QUAL (Base Quality)<30.0, SOR (Strand Odds Ratio)>4.0, FS (Fisher Strand)>60.0, MQ (RMS Mapping Quality)<40.0, MQRankSum (Mapping Quality Rank Sum Test)<–12.5, ReadPosRankSum (Read Pos Rank Sum Test)<–8.0, following the standard in the literature [18]. Mutations were distributed across all eight chromosomes with a normal distribution of adjacent site distances (S4 Fig).

## Split sample processing to estimate sampling error

To estimate technical variation resulting from sample processing and base calling [22,27], we split twenty micropunched samples of Flower #2 just after addition of stop solution during extraction. These paired subsamples were then genome sequenced independently alongside the other samples processed without splitting (Fig 2A). Mutations induced by commercial DNA polymerase or reagents in the Qiagen REPLI-g single-cell kit would be expected to appear in only one of the two subsamples. Thus, any *de novo* mutation called qualitatively in both subsamples is reasonably interpreted as a true positive, whereas a mutation that only occurs in one of the two subsamples is a false positive if not supported by an

additional sample elsewhere in the flower or a false negative if it is absent in one of the subsamples but present in the other subsample of the pair and present in other samples from the flower. Qualitative analysis consisted of compiling relative frequencies of true and false positives and true and false negatives (S5 Table). Quantitative analysis consisted of comparison of VAF (%) estimates (S6 Table) with fits evaluated by orthogonal regression.

### Sector analysis through clustering

Clusters of samples with shared mutations were identified using K means clustering in Minitab 20 and dendrogram neighbor-joining in MEGA7 [49] using the 15 *de novo* floral mutations that occurred in two or more organs (S4 Table). Initials used for K means clustering are indicated in S4 Table. Bootstrap support for the dendrogram clusters was determined from 1000 iterations. Comparison of shared mutations between petals, sepals, and stamens were tested relative to a null expectation of random distribution of 13 events by Chi Square test (df = 2).

### Variant allele frequency (VAF) and cell variant frequency (CVF) estimation

Variant allele frequency (VAF%) was calculated as 100*(#ALT reads/total reads) using all 150mer reads that overlay on the base position of that point mutation in each sample. Variant allele frequency (VAF) can then be doubled to estimate cell variant frequency (CVF), under the following five assumptions: 1) diploidy, 2) normal mitotic cell division, 3) stable cell ploidy, 4) no gene conversion or extrachromosomal duplication at the mutation site, and 5) mutation arises and is retained in the heterozygous state. The first assumption is general knowledge for peach [23]. The second and third assumption of normal mitotic cell division and stable cell ploidy were extensively documented for peach branch and flower tissues in the classic histological studies [6,15]. Violation of the fourth assumption would result in abnormal read count distributions at the mutation site and thus such sites would be identified and filtered out in the mutation identification pipeline (S3 Fig). The fifth assumption of retained heterozygosity is a reasonable assumption given that mitotic cell division does not typically involve interactions between homologous chromosomes, although exceptions have been found in some monocots [50]. Empirically, the assumption of retained heterozygosity is supported in *Prunus*, where Goel and colleagues recently showed that 96% of their focal somatic mutations in an apricot tree were retained in this heterozygous condition through several branch initiation events [35]. Thus, overall these five general conditions are reasonably met in peach, and the approach may well be applicable to many other angiosperms. The important benefit of this cell lineage estimation from next-gen sequencing is that it provides an independent estimate of cell frequencies relative to the histological approaches and may identify cryptic patterns that would not be distinguishable by more traditional methods. In human studies, this "next-gen" cell lineage estimation has yielded insights into cancer tumor formation recently [45]. For quantitative analyses using these cell lineage estimates (S9 Fig), the simplest situation is if there are only two cell lineages involved. More complex situations with three or more genetically distinct lineages are beyond the scope of this paper.

### Histological cross-sections measured for L1 and L2 cell proportions

To quantify the relative proportion of epidermal and interior cells in peach tissues, we used counts from histology cross-sections through leaf, carpel, sepal, stamen, and petal cross-sections from published images (S10 and S11 Figs). Images were included from the following published papers on peach as follows: three independent leaf blades from Fig 2 of Rios et al (2016) [51], sepals from three different papers [6,52,53], petals from two sources [6,8], stamens from three sources [6,52,54], and three carpels from one source [6]. All images were longitudinal sections, apart from the leaf blades which were transverse sections. Tracings were made of each image, and cells were then counted in ImageJ 1.54 with settings: 8-bit and circularity > 0.3, with counts performed separately for the interior cells and the single outer layer of epidermal cells. Because Dermen and colleagues concluded that L2 and L3 are derived from a single underlying cell lineage, we grouped them together as the interior cell count and we focused on their combined percentage relative to L1 cells. Our counts indicated that epidermal (L1) cell proportions (average +/- SE) were highest in petal tissues (25.1% +/- 2.1%),

followed by stamen (14.6% +/- 1.5%) and sepal (14.0% +/- 0.8%), leaf blade (12.2% +/- 1.7%), and carpel (9.4% +/- 1.1%), which has the greatest structural complexity and at least 28–34 total cell layers in the bud cross-sections [6]. Cell counts of epidermis and interior were then summed and then percentages of each were calculated (S9 Table). Correspondence of histology cross-section counts with the molecular data on cell abundances was assessed by linear regression.

**PCR, Sanger sequencing, and Indigo quantification**

To obtain independent quantification of mutation frequencies, we performed PCR, Sanger sequencing, and used the Indigo tool www.gear-genomics.com/indigo/ to integrate chromatogram peak areas of the reference base and the alternative base for each of the four major branch mutations, #M101 and #M119 (Group 1) and #M71 and #M120 (Group 2). We performed these tests on additional plant material from Branch B1, including three independent samples from carpel, sepal, petal, and stamens of three new flowers (Flowers #3–5) as well as replicate DNA samples collected from subtending leaf (1-1-8-7-10-L2) near Flower #2. Primers for #M101(385 bp: CTGTAAAATTTGCGCAAATTG (F) and GTTGTGC-GTGCAGGAAG (R)).

Primers for #M119 (380 bp: GCTAGCTCCTGATAGCATTTGTG (F) and TAAATACCAGCATGGCACCAC (R)). Primers for #M71 (Set#1(496 bp): TGGCTACATAGATACAAGCCC (F) and GATATTCTGATGACTGCCTCCC (R); Set#2(679 bp): TGGCTACATAGATACAAGCCC (F) and GTCGTCTTATCCTAGCAGTGAG (R)). Primers for #M120 (467 bp: GAGTCAT-GAAGGTTCGAACT(F) and GGGTTATTATTGGGGGAGC (R)). Results were analyzed by one-way or two-way ANOVA (S12A–S12E Fig and S10 Table).

**Screening of progeny for Group 1 and Group 2 lineage markers**

Seven seeds collected from Branch B1 in 2019 were grown to saplings in 2020 and the first true leaves screened for Group 1 and Group 2 marking mutations. No control was made over pollen source, and saplings represented a mixture of outcrossed and selfed progeny. If the saplings contain mutations #M120 or #M103, then the Group 2 lineage contributed genetically to these progeny, whereas if the saplings contain mutation #M101, then the Group 1 lineage contributed genetically to these progeny (S12F Fig).

## Supporting information

**S1 Fig. Peach floral structure showing sepal, petal, stamen, and carpel connections.**
(PDF)

**S2 Fig. Peach flower microsampling and split subsampling.**
(PDF)

**S3 Fig. Flowchart of sampling and genome-wide screening of candidates.**
(PDF)

**S4 Fig. Distribution of point mutations on the eight chromosomes.**
(PDF)

**S5 Fig. Distribution of the 57 *de novo* floral mutations (#M01 - #M57) among flower parts.**
(PDF)

**S6 Fig. Analysis of paired split subsamples.**
(PDF)

**S7 Fig. Negative VAF correlations for Group 1 and Group 2 marker pairs.**
(PDF)

**S8 Fig. Positive VAF correlations for Group 1 and Group 2 marker pairs.**
(PDF)

**S9 Fig. Estimation of cell variant frequency (CVF) from variant allele frequency (VAF).**
(PDF)

**S10 Fig. VAF estimates correspond to epidermal and interior cell populations.**
(PDF)

**S11 Fig. Epidermal and interior cell counts in *Prunus persica*.**
(PDF)

**S12 Fig. Group 1 and Group 2 mutations in additional flowers, leaves, and progeny.**
(PDF)

**S13 Fig. Floral bud cross-section chronology.**
(PDF)

**S14 Fig. Effect of endopolyploidy on read-based estimates of cell variant frequency.**
(PDF)

**S1 Table. Mapping depth and coverage information for all genome samples.**
(XLSX)

**S2 Table. Genomic locations of all *de novo* point mutations.**
(XLSX)

**S3 Table. Read counts and base calls in all floral and subtending leaf samples.**
(XLSX)

**S4 Table. Floral sectors marked by shared mutations.**
(XLSX)

**S5 Table. Split sample qualitative comparison of mutation calls.**
(XLSX)

**S6 Table. Split sample quantitative comparison of VAF (%) estimates.**
(XLSX)

**S7 Table. VAF (%) estimates of major Group 1 and Group 2 mutations in all samples.**
(XLSX)

**S8 Table. Cell variant frequency sums approximate 100% in all four marker pairs.**
(XLSX)

**S9 Table. Histology cell counts of epidermis (L1) and interior (L2 derived) proportions.**
(XLSX)

**S10 Table. PCR amplification of #M119, #M101, #M71, and #M120.**
(XLSX)

**S11 Table. Transmission of Group 2 mutations into seven progeny.**
(XLSX)

**S12 Table. Sector Mutations assigned to Group 1 and Group 2.**
(XLSX)

**S13 Table. Genome-wide assessments of somatic mutations in woody flowering plants.**
(XLSX)

## Author contributions

**Conceptualization:** Dacheng Tian, Sihai Yang, Long Wang, Milton Brian Traw, Ju Huang.

**Data curation:** Yilun Ji, Milton Brian Traw, Ju Huang.

**Formal analysis:** Milton Brian Traw.

**Funding acquisition:** Dacheng Tian, Sihai Yang, Long Wang, Ju Huang.

**Investigation:** Yilun Ji, Xiaonan Chen, Xiaohui Zhang, Wenjing Wang, Lan Xue, Yifan Zhong, Long Wang, Milton Brian Traw.

**Methodology:** Xiaohui Zhang, Dacheng Tian, Sihai Yang, Long Wang, Milton Brian Traw, Ju Huang.

**Project administration:** Dacheng Tian, Sihai Yang, Milton Brian Traw, Ju Huang.

**Software:** Ju Huang.

**Supervision:** Long Wang, Milton Brian Traw, Ju Huang.

**Validation:** Yilun Ji, Xiaonan Chen, Xiaohui Zhang, Wenjing Wang.

**Visualization:** Yilun Ji, Wenjing Wang, Milton Brian Traw.

**Writing – original draft:** Milton Brian Traw.

**Writing – review & editing:** Sihai Yang, Long Wang, Milton Brian Traw, Ju Huang.

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
