## [Decision Letter · Decision Letter 0]

26 Apr 2025

PGENETICS-D-25-00173

Mutations mark cell lineages and sectors in flowers of a woody angiosperm

PLOS Genetics

Dear Dr. Traw,

Thank you for submitting your manuscript to PLOS Genetics. After careful consideration, we feel that it has merit but does not fully meet PLOS Genetics's publication criteria as it currently stands. Therefore, we invite you to submit a revised version of the manuscript that addresses the points raised during the review process.

Please submit your revised manuscript within 60 days Jun 25 2025 11:59PM. If you will need more time than this to complete your revisions, please reply to this message or contact the journal office at plosgenetics@plos.org. Please include the following items when submitting your revised manuscript:

We look forward to receiving your revised manuscript.

Kind regards,

Anne Goriely

Editor-in-Chief

PLOS Genetics

Anne Goriely

Editor-in-Chief

PLOS Genetics

Aimée Dudley

Editor-in-Chief

PLOS Genetics

Anne Goriely

Editor-in-Chief

PLOS Genetics

**Additional Editor Comments (if provided):**

**Journal Requirements:**

**Reviewers' comments:**

Reviewer's Responses to Questions

**Comments to the Authors:**

Reviewer #1: This study investigates floral development in peach by tracking genome-wide de novo mutations in two individual flowers. Specifically, the authors ask whether shared mutations define radial sectors that span sepals or sepals and adjacent petals, whether carpels share mutations with radial sectors, and whether mutation patterns can be used to estimate the number of founder cells in the floral anlagen. The results are intriguing: the authors found that certain mutations are shared between stamens and sepals, but not petals, and that each flower harbored at least two mutations spanning the entire floral structure. These findings are potentially significant. However, the methodology, data presentation, and interpretation require clarification.

A major concern lies in the interpretation of SNVs observed at the cell population level, particularly in the context of cell lineage. Since the study does not clearly state what type of stem cell dynamics or lineage behavior is assumed, it is difficult to assess the implications of the observed mutation patterns. The authors should either clarify their assumptions regarding cell lineage and developmental dynamics, or incorporate a mathematical model to bridge the gap between population-level mutation data and cellular-level developmental processes.

The following are more specific comments:

Lines 103–108:

It seems that the time elapsed since the bifurcation of flowers 1 and 2 from the stem cell population is relatively short—likely several weeks or months. Given that 70 de novo point mutations were detected, what would this imply for the de novo mutation rate per year? Please provide an estimate of the mutation rate based on your data and compare it with previously reported rates in plants or other systems.

Lines 153–157:

This is an intriguing result. However, the interpretation that a negative correlation between different de novo mutations supports the existence of two underlying cell lineages in each flower is not fully convincing. Please clarify the logic behind this conclusion. Specifically, explain how such a correlation pattern would arise from two distinct lineages and whether alternative explanations were considered.

Lines 159–170:

The authors present an interesting argument; however, it is difficult to draw definitive conclusions from data at the cell population level alone. Stem cells may be lost due to drift, which complicates lineage inference. To support their interpretation, the authors should consider the underlying stem cell population dynamics and discuss what types of dynamics could account for the observed mutation patterns. Incorporating a mathematical model could help bridge the gap between individual cellular behavior and population-level observations (e.g., Satake and Tomimoto, Trends in Genetics, 2024: https://www.sciencedirect.com/science/article/pii/S0168952524001070).

In addition, the resolution of Fig. 3A is low and should be improved for clarity.

Line 171:

Regarding cell abundance in L1 and L2 layers, please provide actual microscopic images, rather than illustrations, used in the estimation. Authors used published images, but environmental conditions are different, probably accession too. I suggest to collect images from the same individual used in this study. The method used to estimate cell abundances also needs to be clearly described.

Line 352:

Since GATK’s UnifiedGenotyper is known to yield a relatively high false positive rate, the reliability of SNV calls should be validated using additional tools such as GATK HaplotypeCaller, which is more commonly used in recent studies. In Fig. S3, it appears that both UnifiedGenotyper and HaplotypeCaller were used. Please clarify this point.

Line 354:

More specific filtering criteria for visual inspection of SNVs should be described. Fig. S3, which appears relevant, should be cited here.

Figure 2B, C:

Please explain what "a" and "b" represent—presumably replicates? If so, please state this explicitly.

Figure 2D, E:

Y-axis labels are missing in the bar plots. Please add them to improve readability.

Data availability

It seems like accession number is still not obtained.

I also think that authors need to provide code used for analyses.

Reviewer #2: This study by Ji et al. is one of the first attempts to describe which cell lineages lead to the development of which plant sub-part, following the accumulation of somatic mutations in a woody plant (peach). This approach complements and deepens the knowledge we have from histological studies, which suggests a two cell bottleneck at the onset of flower development. Using next-generation sequencing, the study examines the accumulation of nested mutations, offers valuable insights into the early segregation of cell lineages in flowers and lays the groundwork for further genomic explorations in plant developmental biology.

The authors collected micro-punch samples from two flowers and performed whole-genome sequencing to an average depth of 64.3x. They identified a total of 70 mutations: 20 in Flower #1 and 13 plus 37 in Flower #2. By analyzing mutations shared among different samples, they were able to investigate the phenomenon of sectoring.

To evaluate the reliability of their somatic mutation calls, the authors split the samples into two batches, generating independent variant calls of the same genome. Their high replication rate for somatic mutations (97%) confirms that these mutations can be effectively tracked in the different flower parts. They also verified that the percent variant allele frequency (VAF%) was consistent between independent mutation calls.

They find, by focusing on two mutations present flower-wide and whose frequency is negatively correlated with each other, evidence in support of two underlying cell lineages giving rise to a flower. This is further confirmed by comparing mutations found in earlier (branch) tissues. Specifically, they identify branch-derived mutations present in both flowers, some of which correlate positively and others negatively with the flower-specific mutations. Based on these observations, the authors assign the mutations to two separate groups corresponding to two distinct cell lineages.

To test whether each flower’s component cell lineages emerged from a single founder cell, the authors compared the VAF% in major tree branches to those in the flowers. In three out of four cases, the relative abundances of the groups’ mutations remained the same in the main tree and the flower, suggesting that frequencies did not change over the transition from branch to flower. When they plotted the frequency of these mutations over time (as the tree grew), they found that the proportion of group 2 cells tended to decrease.

Finally, in Figure 5, they recapitulate the cell lineages contributions to flower development by following the emergence of mutations in the different sectors that they describe in Figure 1.

Overall summary :

I find this paper clever and relevant - an important contribution to the field of plant development and molecular biology. Their approach is clearly illustrated with figures and schemas, although some of them could be clearer and simpler (see comments below).

Minor comments

L5: Remove redundant "whether" in "ask whether 1) whether radial sectors accumulate nested mutations."

L87-89 : the sampling and subsampling design has not been described and the term Flower #2 cannot be understood at that stage. Consider re-phrasing/giving more detail at this stage of the paper.

L104 : authors find a total of 70 new mutations between two flowers - this is a very high number of mutations. How does it compare to the number of mutations we would expect in annual tissues ? I believe this is missing in the discussion.

Fig 1. C&D : the labels are too small to be read/informative.

Figure S3 filter criteria - very good chart. Are the filter criteria all exclusive ? i.e. if they meet the filter, they get filtered out ? I think that could be a bit more clear in the legend.

Mutation #M42 is used as a focal/example mutation for Figure 2B, but what is the justification behind this choice ?

Fig. 3. : The visual could be improved : the table and labels are blurry and the information could be conveyed in a clearer way with a bar plot for example. Overall, the quality of graphics is not consistent, make sure to export vector format figures to ensure high resolution.

Fig. 4. and L205 : can you explain a bit more how and why you choose the thresholds for Group 1 versus Group 2 mutations ?

Reviewer #3: Ji and colleagues use high-coverage short read sequencing to explore patterns of somatic mutations in two peach flowers. The importance of characterizing somatic mutations in floral tissue can be related to 1) understanding the number of founder cells that give rise to the developed flower and 2) determining to what extent somatic mutations are transmitted to progeny. The authors present a very nice data set to explore these questions using both in-depth sampling of floral tissue and high depth of sequencing to uncover somatic mutations that may occur at low variant allele frequencies. The main findings of this work are that 1) somatic mutations are nested and can be traced to determine cell lineages, 2) that there are likely only two floral founder cells (one gives rise to L1 and one to L2), and 3) carpels do not share these mutations with the rest of the floral organs. Overall, the dataset and results presented are exciting and of interest of researchers in the field. I noted a few concerns below.

1) The results presented in the manuscript depend heavily on the SNP calling methodology. I was disappointed that methods used for SNP calling only referenced “standard guidelines for screening somatic point mutations” (Barnell, 2019). This referenced article outlines a workflow for calling somatic mutations between tumor/normal sample pairs. If such a paired strategy was used then it must be more carefully detailed in the methods (i.e. what was the “normal” sample?). Similarly, Figure S3 outlines criteria 4 for filtering and some additional clarifications are needed. Criteria 4) SNV is parental heterozygosity (in all samples). I understood the genome was diploid. If mapping to a diploid genome then parental heterozygozity should be reflected as differences between the haplotypes (not variants identified via mapping of short reads to the reference). Is the reference genome assembled from the same tree? Please provide clarification. Criteria 7/8) 3 or more SNV reads across all controls. 2 or more SNV reads in any single control. What were the control samples? Please clarify. Most importantly, Why were 680 of 750 (80%) SNVs remaining after filtering based on criteria 1-11 excluded by visual check? What were the criteria for exclusion? These 680 variants were characterized as true negatives in S6A. How was this determined? While the results from the small number of high confidence variants (70) were intriguing. I am very concerned about the exclusion of 80% of variants at this stage.

2) One of the key findings of the manuscript is that there are two cell lineages marked by somatic variants in each flower. Convincing evidence was presented to suggest that there is likely just two progenitor cells, one each for the L1 and L2 layers and that the variant frequency for mutations in each lineage are anticorrelated. The lack of shared somatic mutations in the carpels was concerning. The authors provide some argument that carpels are distinct, with their own vasculature, and so this could be an explanation for their observations. Please provide additional supporting evidence or references to support the hypothesis that the carpel cell lineage is not expected to derive from one of the two floral founder cells.

3) The authors also observe that carpel cell lineages do not share mutations with the rest of the floral organs and hypothesize that female gametes are protected from mutation. What about the male gametes? Is there evidence to support reduced mutation in female vs male gametes in peach? Or other tree crops?

4) Only L2 associated mutations were identified in surveyed progeny. This is evidence that some somatic mutations can be transmitted. I was surprised that these mutations were not fixed in the progeny and occurred at a range of variant allele frequencies. Variants transmitted through the egg cell or sperm cell should be fixed (or at a VAF of 0.5 for haploid reference genomes).

**Have all data underlying the figures and results presented in the manuscript been provided?**

Reviewer #1: Yes

Reviewer #2: Yes

Reviewer #3: **No: ** Not clear if the underlying data has been uploaded to NCBI SRA

PLOS authors have the option to publish the peer review history of their article (what does this mean? ). If published, this will include your full peer review and any attached files.

**Do you want your identity to be public for this peer review?** For information about this choice, including consent withdrawal, please see our Privacy Policy .

Reviewer #1: No

Reviewer #2: No

Reviewer #3: No

**Figure resubmission:**
---

## [Decision Letter · Decision Letter 1]

3 Aug 2025

Dear Dr Traw,

We are pleased to inform you that your manuscript entitled "Mutations mark cell lineages and sectors in flowers of a woody angiosperm" has been editorially accepted for publication in PLOS Genetics. As you can see below all the three reviewers are satisfied with the changes made to the original submission.Congratulations! 

Yours sincerely,

Arun Sampathkumar

Academic Editor

PLOS Genetics

Anne Goriely

Editor-in-Chief

PLOS Genetics

Aimée Dudley

Editor-in-Chief

PLOS Genetics

Anne Goriely

Editor-in-Chief

PLOS Genetics

Comments from the reviewers (if applicable):

Reviewer's Responses to Questions

**Comments to the Authors:**

Reviewer #1: This manuscript is intensively revised, and it is now ready for publication.

Reviewer #2: The authors have addressed my comments and I am satisfied with the changes they made to the manuscript. I also overall agree with the other reviewers' comments and suggestions and I approve this manuscript for publication. I believe it is a good step towards a better understanding of plant growth, cell division within plant parts and mutational dynamics within cell lineages.

Reviewer #3: The authors sufficiently addressed the comments and concerns in the revised manuscript.

**Have all data underlying the figures and results presented in the manuscript been provided?**

Reviewer #1: None

Reviewer #2: Yes

Reviewer #3: **No: ** SRA accession ID is provided, but link was not active.

PLOS authors have the option to publish the peer review history of their article (what does this mean? ). If published, this will include your full peer review and any attached files.

**Do you want your identity to be public for this peer review?** For information about this choice, including consent withdrawal, please see our Privacy Policy .

Reviewer #1: No

Reviewer #2: No

Reviewer #3: No

**Data Deposition**

http://datadryad.org/submit?journalID=pgenetics&manu=PGENETICS-D-25-00173R1

**Press Queries**

---

## [Editor Report · Acceptance letter]

PGENETICS-D-25-00173R1

Mutations mark cell lineages and sectors in flowers of a woody angiosperm

Dear Dr Traw,

We are pleased to inform you that your manuscript entitled "Mutations mark cell lineages and sectors in flowers of a woody angiosperm" has been formally accepted for publication in PLOS Genetics! Your manuscript is now with our production department and you will be notified of the publication date in due course.

With kind regards,

Zsofia Freund

PLOS Genetics

On behalf of:
